# Communication Efficient Parallel Algorithms for Optimization on Manifolds

**Bayan Saparbayeva**
Department of Applied and
Computational Mathematics and Statistics
University of Notre Dame
Notre Dame, Indiana 46556, USA
`bsaparba@nd.edu`

**Michael Minyi Zhang**
Department of Computer Science
Princeton University
Princeton, New Jersey 08540, USA
`mz8@cs.princeton.edu`

**Lizhen Lin**
Department of Applied and
Computational Mathematics and Statistics
University of Notre Dame
Notre Dame, Indiana 46556, USA
`lizhen.lin@nd.edu`

## Abstract

The last decade has witnessed an explosion in the development of models, theory and computational algorithms for "big data" analysis. In particular, distributed computing has served as a natural and dominating paradigm for statistical inference. However, the existing literature on parallel inference almost exclusively focuses on Euclidean data and parameters. While this assumption is valid for many applications, it is increasingly more common to encounter problems where the data or the parameters lie on a non-Euclidean space, like a manifold for example. Our work aims to fill a critical gap in the literature by generalizing parallel inference algorithms to optimization on manifolds. We show that our proposed algorithm is both communication efficient and carries theoretical convergence guarantees. In addition, we demonstrate the performance of our algorithm to the estimation of Fréchet means on simulated spherical data and the low-rank matrix completion problem over Grassmann manifolds applied to the Netflix prize data set.

## 1   Introduction

A natural representation for many statistical and machine learning problems is to assume the parameter of interest lies on a more general space than the Euclidean space. Typical examples of this situation include diffusion matrices in large scale diffusion tensor imaging (DTI) which are $3 \times 3$ positive definite matrices, now commonly used in neuroimaging for clinical trials [1]. In computer vision, images are often preprocessed or reduced to a collection of subspaces [11, 27] or, a digital image can also be represented by a set of $k$-landmarks, forming landmark based shapes [13]. One may also encounter data that are stored as orthonormal frames [8], surfaces[15], curves[16], and networks [14].

In addition, parallel inference has become popular in overcoming the computational burden arising from the storage, processing and computation of big data, resulting in a vast literature in statistics and machine learning dedicated to this topic. The general scheme in the frequentist setting is to divide the data into subsets, obtain estimates from each subset which are combined to form an ultimate estimate for inference [9, 30, 17]. In the Bayesian setting, the subset posterior distributions are first obtained in the dividing step, and these subset posterior measures or the MCMC samples from each subset

posterior are then combined for final inference [20, 29, 28, 21, 25, 22]. Most of these methods are "embarrassingly parallel" which often do not require communication across different machines or subsets. Some communication efficient algorithms have also been proposed with prominent methods including [12] and [26].

Despite tremendous advancement in parallel inference, previous work largely focuses only on Euclidean data and parameter spaces. To better address challenges arising from inference of big non-Euclidean data or data with non-Euclidean parameters, there is a crucial need for developing valid and efficient inference methods including parallel or distributed inference and algorithms that can appropriately incorporate the underlying geometric structure.

For a majority of applications, the parameter spaces fall into the general category of *manifolds*, whose geometry is well-characterized. Although there is a recent literature on inference of manifold-valued data including methods based on Fréchet means or model based methods [3, 4, 5, 2, 18] and even scalable methods for certain models [23, 19, 24], there is still a vital lack of general parallel algorithms on manifolds. We aim to fill this critical gap by introducing our parallel inference strategy. The novelty of our paper is in the fact that is generalizable to a wide range of loss functions for manifold optimization problems and that we can parallelize the algorithm by splitting the data across processors. Furthermore, our theoretical development does not rely on previous results. In fact, generalizing Theorem 1 to the manifold setting requires totally different machineries from that of previous work.

Notably, our parallel optimization algorithm has several key features:

(1) Our parallel algorithm efficiently exploits the geometric information of the data or parameters.

(2) The algorithm minimizes expensive inter-processor communication.

(3) The algorithm has theoretical guarantees in approximating the true optimizer, characterized in terms of convergence rates.

(4) The algorithm has outstanding practical performance in simulation studies and real data examples.

Our paper is organized as follows: In Section 2 we introduce related work to the topic of parallel inference. Next we present our proposed parallel optimization framework in Section 3 and present theoretical convergence results for our parallel algorithm in Section 4. In Section 5, we consider a simulation study of estimating the Fréchet means on the spheres and a real data example using the Netflix prize data set. The paper ends with a conclusion and discussion of future work in Section 6.

## 2 Related work

In the typical "big data" scenario, it is usually the case that the entire data set cannot fit onto one machine. Hence, parallel inference algorithms with provably good theoretic convergence properties are crucial for this situation. In such a setting, we assume that we have $N = mn$ identically distributed observations $\{x_{ij} : i = 1, \ldots, n, j = 1, \ldots, m\}$, which are i.i.d divided into $m$ subsets $X_j = \{x_{ij}, i = 1, \ldots, n\}, j = 1, \ldots, m$ and stored in $m$ separate machines. While it is important to consider inference problems when the data are not i.i.d. distributed across processors, we will only consider the i.i.d. setting as a simplifying assumption for the theory.

For a loss function $\mathscr{L} : \Theta \times \mathscr{D} \to \mathbb{R}$, each machine $j$ has access to a local loss function, $\mathscr{L}_j(\theta) = \frac{1}{n} \sum_{i=1}^{n} \mathscr{L}(\theta, x_{ij})$, where $\mathscr{D}$ is the data space. Then, the local loss functions are combined into a global loss function $\mathscr{L}_N(\theta) = \frac{1}{m} \sum_{j=1}^{m} \mathscr{L}_j(\theta)$. For our intended optimization routine, we are actually looking for the minimizer of an expected loss function $\mathscr{L}^*(\theta) = \mathbb{E}_{x \in \mathscr{D}} \mathscr{L}(\theta, x)$. In the parallel setting, we cannot investigate $\mathscr{L}^*$ directly and we may only analyze it through $\mathscr{L}_N$. However, calculating the total loss function directly and exactly requires excessive inter-processor communication, which carries a huge computational burden as the number of processors increase. Thus, we must approximate the true parameter $\theta^* = \operatorname{argmin}_{\theta \in \Theta} \mathscr{L}^*(\theta)$ by an empirical risk minimizer $\hat{\theta} = \operatorname{argmin}_{\theta \in \Theta} \mathscr{L}_N(\theta)$.

In this work, we focus on generalizing a particular parallel inference framework, the Iterative Local Estimation Algorithm (ILEA) [12], to manifolds. This algorithm optimizes an approximate, surrogate loss function instead of the global loss function as a way to avoid processor communication. The

idea of the surrogate function starts from the Taylor series expansion of $\mathscr{L}_N$

$$\mathscr{L}_N\big(\bar\theta + t(\theta - \bar\theta)\big) = \mathscr{L}_N(\bar\theta) + t\langle\nabla\mathscr{L}_N(\bar\theta), \theta - \bar\theta\rangle + \sum_{s=2}^{\infty} \frac{t^s}{s!}\nabla^s\mathscr{L}_N(\bar\theta)(\theta - \bar\theta)^{\otimes s}.$$

The global high-order derivatives $\nabla^s\mathscr{L}_N(\bar\theta)$ $(s \geq 2)$ are replaced by local high-order derivatives $\nabla^s\mathscr{L}_1(\bar\theta)(s \geq 2)$ from the first machine

$$\tilde{\mathscr{L}}(\theta) = \mathscr{L}_N(\bar\theta) + \langle\nabla\mathscr{L}_N(\bar\theta), \theta - \bar\theta\rangle + \sum_{s=2}^{\infty} \frac{1}{s!}\nabla^s\mathscr{L}_1(\bar\theta)(\theta - \bar\theta)^{\otimes s}.$$

So the approximation error is

$$\begin{aligned}
\tilde{\mathscr{L}}(\theta) - \mathscr{L}_N(\theta) &= \sum_{s=2}^{\infty} \frac{1}{s!}\big(\nabla^s\mathscr{L}_1(\bar\theta) - \nabla^s\mathscr{L}_N(\bar\theta)\big)(\theta - \bar\theta)^{\otimes s}\\
&= \frac{1}{2}\Big\langle\theta - \bar\theta, \big(\nabla^2\mathscr{L}_1(\bar\theta) - \nabla^2\mathscr{L}_N(\bar\theta)\big)(\theta - \bar\theta)\Big\rangle + O\big(\|\theta - \bar\theta\|^3\big)\\
&= O\Big(\frac{1}{\sqrt{n}}\|\theta - \bar\theta\|^2 + \|\theta - \bar\theta\|^3\Big).
\end{aligned}$$

The infinite sum $\sum_{s=2}^{\infty}\frac{1}{s!}\nabla^s\mathscr{L}_1(\bar\theta)(\theta - \bar\theta)^{\otimes s}$ in the $\tilde{\mathscr{L}}(\theta)$ can be replaced by $\mathscr{L}_1(\theta) - \mathscr{L}_1(\bar\theta) - \langle\nabla\mathscr{L}_1(\bar\theta), \theta - \bar\theta\rangle$

$$\tilde{\mathscr{L}}(\theta) = \mathscr{L}_1(\theta) - \big(\mathscr{L}_1(\bar\theta) - \mathscr{L}_N(\bar\theta)\big) - \big\langle\nabla\mathscr{L}_1(\bar\theta) - \nabla\mathscr{L}_N(\bar\theta), \theta - \bar\theta\big\rangle.$$

We can omit the additive constant $\big(\mathscr{L}_1(\bar\theta) - \mathscr{L}_N(\bar\theta)\big) + \langle\nabla\mathscr{L}_1(\bar\theta) - \nabla\mathscr{L}_N(\bar\theta), \bar\theta\rangle$. Thus the surrogate loss function $\tilde{\mathscr{L}}(\theta)$ is defined as

$$\tilde{\mathscr{L}}(\theta) = \mathscr{L}_1(\theta) - \langle\nabla\mathscr{L}_1(\bar\theta) - \nabla\mathscr{L}_N(\bar\theta), \theta\rangle.$$

Thus, the surrogate minimizer $\tilde\theta = \text{argmin}_\Theta \tilde{\mathscr{L}}$ approximates the empirical risk minimizer $\hat\theta$.

[12] show that the consequent surrogate minimizers have a provably good convergence rate to $\hat\theta$ given the following regularity conditions:

1. The parameter space $\Theta$ is a compact and convex subset of $\mathbb{R}^d$. Besides, $\theta^* \in \text{int}(\Theta)$ and $R = \sup_{\theta\in\Theta}\|\theta - \theta^*\| > 0$,

2. The Hessian matrix $I(\theta) = \nabla^2\mathscr{L}^*(\theta)$ is invertible at $\theta^*$, that is there exist constants $(\mu_-, \mu_+)$ such that

$$\mu_- I_d \preceq I(\theta^*) \preceq \mu_+ I_d,$$

3. For any $\delta > 0$, there exists $\epsilon > 0$, such that

$$\inf \mathbb{P}\Big\{\inf_{\|\theta - \theta^*\| \geq \delta}\big|\mathscr{L}(\theta) - \mathscr{L}(\theta^*)\big| \geq \epsilon\Big\} = 1,$$

4. For a ball around the true parameter $U(\rho) = \{\theta : \|\theta - \theta^*\| \leq \rho\}$ there exist constants $(G, L)$ and a function $\mathscr{K}(x)$ such that

$$\mathbb{E}\|\nabla\mathscr{L}(\theta)\|^{16} \leq G^{16} \quad \mathbb{E}\|\|\nabla^2\mathscr{L}(\theta) - I(\theta)\|\| \leq L^{16},$$
$$\|\|\mathscr{L}(\theta, x) - \mathscr{L}(\theta', x)\|\| \leq \mathscr{K}(x)\|\theta - \theta'\|,$$

for all $\theta, \theta' \in U(\rho)$.

which leads to the following theorem:

**Theorem 1.** Suppose that the standard regularity conditions hold and initial estimator $\bar\theta$ lies in the neighborhood $U(\rho)$ of $\theta^*$. Then the minimizer $\tilde\theta$ of the surrogate loss function $\tilde{\mathscr{L}}(\theta)$ satisfies

$$\|\tilde\theta - \hat\theta\| \leq C_2(\|\bar\theta - \hat\theta\| + \|\hat\theta - \theta^*\| + \|\|\nabla^2\mathscr{L}_1(\theta^*) - \nabla^2\mathscr{L}_N(\theta^*)\|\|)\|\bar\theta - \hat\theta\|,$$

with probability at least $1 - C_1 mn^{-8}$, where the constants $C_1$ and $C_2$ are independent of $(m, n, N)$.

# 3   Parallel optimizations on manifolds

Our work aims to generalize the typical gradient descent optimization framework to manifold optimization. In particular, we will use the ILEA framework as our working example to generalize parallel optimization algorithms. Instead of working with $\mathbb{R}^d$, we have a $d$-dimensional manifold $M$. We also consider a surrogate loss function $\tilde{\mathscr{L}}_j : \Theta \times \mathcal{I} \to \mathbb{R}$, where $\Theta$ is a subset of the manifold $M$, that approximates the global loss function $\mathscr{L}_N$. Here we choose to optimize $\tilde{\mathscr{L}}_j$ on the $j$th machine–that is, on different iterations we optimize on different machine for efficient exploration unlike from previous algorithm, where the surrogate function is always optimized on the first machine.

To generalize the idea of moving along a gradient on the manifold $M$, we use the retraction map, which is not necessarily the exponential map that one would typically use in manifold gradient descent, but shares several important properties with the exponential map. Namely, a retraction on $M$ is a smooth mapping $\mathscr{R} : TM \to M$ with the following properties

1. $\mathscr{R}_\theta(0_\theta) = \mathscr{R}(\theta, 0_\theta) = \theta$, where $\mathscr{R}_\theta$ is the restriction of $\mathscr{R}$ from $TM$ to the point $\theta$ and the tangent space $T_\theta M$, $0_\theta$ denotes the zero vector on $T_\theta M$,

2. $D\mathscr{R}_\theta(0_\theta) = D\mathscr{R}(\theta, 0_\theta) = \mathrm{id}_{T_\theta M}$, where $\mathrm{id}_{T_\theta M}$ denotes the identity mapping on $T_\theta M$.

We also demand that

1. For any $\theta_1, \theta_2 \in M$, curves $\mathscr{R}_{\theta_1} t \mathscr{R}_{\theta_1}^{-1} \theta_2$ and $\mathscr{R}_{\theta_2} s \mathscr{R}_{\theta_2}^{-1} \theta_1$, where $s, t \in [0, 1]$, must coincide,

2. The triangle inequality holds, that is for any $\theta_1, \theta_2, \theta_3 \in M$, it is the case that $d_{\mathscr{R}}(\theta_1, \theta_2) \leq d_{\mathscr{R}}(\theta_2, \theta_3) + d_{\mathscr{R}}(\theta_3, \theta_1)$ where $d_{\mathscr{R}}(\theta_1, \theta_2)$ is the length of the curve $\mathscr{R}_{\theta_1} t \mathscr{R}_{\theta_1}^{-1} \theta_2$ for $t \in [0, 1]$.

Our construction starts with the Taylor's formula for $\mathscr{L}_N$ on the manifold $M$

$$\mathscr{L}_N(\theta) = \mathscr{L}_N(\bar{\theta}) + \langle \nabla \mathscr{L}_N(\bar{\theta}), \log_{\bar{\theta}} \theta \rangle + \sum_{s=2}^{\infty} \frac{1}{s!} \nabla^s \mathscr{L}_N(\bar{\theta})(\log_{\bar{\theta}} \theta)^{\otimes s}$$

Because we split the data across machines, evaluating the derivatives $\nabla^s \mathscr{L}_N(\bar{\theta})$ requires excessive processor communication. We want to reduce the amount of communication by replacing the *global high-order derivatives* $\nabla^s \mathscr{L}_N(\bar{\theta})$ ($s \geq 2$) with the *high-order local derivatives* $\nabla^s \mathscr{L}_j(\bar{\theta})$. This gives us the following surrogate to $\mathscr{L}_N$

$$\tilde{\mathscr{L}}_j(\theta) = \mathscr{L}_N(\bar{\theta}) + \langle \nabla \mathscr{L}_N(\bar{\theta}), \log_{\bar{\theta}} \theta \rangle + \sum_{s=2}^{\infty} \frac{1}{s!} \nabla^s \mathscr{L}_j(\bar{\theta})(\log_{\bar{\theta}} \theta)^{\otimes s}.$$

Then we have the following approximation error

$$\tilde{\mathscr{L}}_j(\theta) - \mathscr{L}_N(\theta) = \frac{1}{2} \langle \log_{\bar{\theta}} \theta, (\nabla^2 \mathscr{L}_j(\bar{\theta}) - \nabla^2 \mathscr{L}_N(\bar{\theta})) \log_{\bar{\theta}} \theta \rangle + O\big(d_g(\bar{\theta}, \theta)^3\big)$$

$$= O\Big(\frac{1}{\sqrt{n}} d_g(\bar{\theta}, \theta)^2 + d_g(\bar{\theta}, \theta)^3\Big).$$

We replace $\sum_{s=2}^{\infty} \frac{1}{s!} \nabla^s \mathscr{L}_j(\bar{\theta})(\log_{\bar{\theta}} \theta)^{\otimes s}$ with $\mathscr{L}_j(\theta) - \mathscr{L}_j(\bar{\theta}) - \langle \nabla \mathscr{L}_j(\bar{\theta}), \log_{\bar{\theta}} \theta \rangle$:

$$\tilde{\mathscr{L}}_j(\theta) = \mathscr{L}_N(\bar{\theta}) + \langle \nabla \mathscr{L}_N(\bar{\theta}), \log_{\bar{\theta}} \theta \rangle + \mathscr{L}_j(\theta) - \mathscr{L}_j(\bar{\theta}) - \langle \nabla \mathscr{L}_j(\bar{\theta}), \log_{\bar{\theta}} \theta \rangle$$

$$= \mathscr{L}_j(\theta) + (\mathscr{L}_N(\bar{\theta}) - \mathscr{L}_j(\bar{\theta})) + \langle \nabla \mathscr{L}_N(\bar{\theta}) - \nabla \mathscr{L}_j(\bar{\theta}), \log_{\bar{\theta}} \theta \rangle.$$

Since we are not interested in the value of $\tilde{\mathscr{L}}_j$ but in its minimizer, we omit the additive constant $(\mathscr{L}_N(\bar{\theta}) - \mathscr{L}_j(\bar{\theta}))$ and redefine $\tilde{\mathscr{L}}_j$ as $\tilde{\mathscr{L}}_j(\theta) := \mathscr{L}_j(\theta) - \langle \nabla \mathscr{L}_j(\bar{\theta}) - \nabla \mathscr{L}_N(\bar{\theta}), \log_{\bar{\theta}} \theta \rangle$. Then we can generalize the exponential map $\exp_{\bar{\theta}}$ and the inverse exponential map $\log_{\bar{\theta}}$ to the retraction map $\mathscr{R}_{\bar{\theta}}$ and the inverse retraction map $\mathscr{R}_{\bar{\theta}}^{-1}$, which is also called the lifting, and redefine $\tilde{\mathscr{L}}_j$

$$\tilde{\mathscr{L}}_j(\theta) := \mathscr{L}_j(\theta) - \langle \nabla \mathscr{L}_j(\bar{\theta}) - \nabla \mathscr{L}_N(\bar{\theta}), \mathscr{R}_{\bar{\theta}}^{-1} \theta \rangle.$$

Therefore we have the following generalization of the Iterative Local Estimation Algorithm (ILEA) for the manifold $M$:

---
**Algorithm 1:** ILEA for Manifolds
---
Initialize $\theta_0 = \bar{\theta}$;
**for** $s = 0, 1, \ldots, T-1$ **do**

    Transmit the current iterate $\theta_s$ to local machines $\{\mathcal{M}_j\}_{j=1}^m$;

    **for** $j = 1, \ldots, m$ **do**

        Compute the local gradient $\nabla\mathcal{L}_j(\theta_s)$ at machine $\mathcal{M}_j$;

        Transmit the local gradient $\nabla\mathcal{L}_j(\theta_s)$ to machine $\mathcal{M}_s$;

    Calculate the global gradient $\nabla\mathcal{L}_N(\theta_s) = \frac{1}{m}\sum_{j=1}^m \nabla\mathcal{L}_j(\theta_s))$ in Machine $\mathcal{M}_s$;

    Form the surrogate function $\tilde{\mathcal{L}}_s(\theta) = \mathcal{L}_s(\theta) - \langle\mathcal{R}_{\theta_s}^{-1}\theta, \nabla\mathcal{L}_s(\theta_s) - \nabla\mathcal{L}_N(\theta_s)\rangle$;

    Update $\theta_{s+1} \in \arg\min \tilde{\mathcal{L}}_s$;

Return $\theta_T$
---

# 4 Convergence rates of the algorithm

To establish some theoretical convergence rates on our algorithm, we consequently have to impose some regularity conditions on the parameter space $\Theta$, the loss function $\mathcal{L}$ and the population risk $\mathcal{L}^*$. We must establish these conditions specifically for manifolds instead of simply using the regularity conditions placed on Euclidean spaces. For example, in the manifold the Hessians $\nabla^2\mathcal{L}(\theta, x), \nabla^2\mathcal{L}(\theta', x)$ are defined in different tangent spaces meaning there cannot be any linear expressions of the second-order derivatives.

In the manifold for any $\xi \in T_{\theta'}M$ we can define the vector field as $\xi(\theta) = D(\mathcal{R}_\theta^{-1}\theta')\xi$. We can also take the covariant derivative of $\xi(\theta)$ along the retraction $\mathcal{R}_{\theta'}t\mathcal{R}_{\theta'}\theta$:

$$\nabla_{D\left(R_{\theta'}^{-1}(R_{\theta'}t\mathcal{R}_{\theta'}\theta)\right)^{-1}\mathcal{R}_{\theta'}^{-1}\theta}\xi(\mathcal{R}_{\theta'}t\mathcal{R}_{\theta'}\theta) =$$

$$\nabla_{D\left(R_{\theta'}^{-1}(R_{\theta'}t\mathcal{R}_{\theta'}\theta)\right)^{-1}\mathcal{R}_{\theta'}^{-1}\theta}D\left(\mathcal{R}_{\mathcal{R}_{\theta'}t\mathcal{R}_{\theta'}\theta}^{-1}\theta'\right)\xi = \nabla D(t, \theta, \theta')\xi. \quad (1)$$

The expression (1) defines the linear map $\nabla D(t, \theta, \theta')$ from $T_{\theta'}M$ to $T_{\mathcal{R}_{\theta'}t\mathcal{R}_{\theta'}\theta}M$ and want to impose some conditions to this map. Finally, we impose the following regularity conditions on the parameter space $\Theta$, the loss function $\mathcal{L}$ and the population risk $\mathcal{L}^*$.

1. The parameter space $\Theta$ is a compact and $\mathcal{R}$-convex subset of $M$, which means that for any $\theta_1, \theta_2 \in \Theta$ curves $\mathcal{R}_{\theta_1}t\mathcal{R}_{\theta_1}\theta_2$ and $\exp_{\theta_1}t\log_{\theta_1}\theta_2$ must be within $\Theta$ for any $\theta_1, \theta_2 \in M$ and also demand that there exists $L' \in \mathbb{R}$ such that

$$d_{\mathcal{R}}(\theta_1, \theta_2) \le L'd_g(\theta_1, \theta_2),$$

    where $d_g(\theta_1, \theta_2)$ is the geodesic distance,

2. The matrix $I(\theta) = \nabla^2\mathcal{L}^*(\theta)$ is invertible at $\theta^*$ : $\exists$ constants $\mu_-, \mu_+ \in \mathbb{R}$ such that

$$\mu_-\text{id}_{\theta^*} \preceq I(\theta^*) \preceq \mu_+\text{id}_{\theta^*},$$

3. For any $\delta > 0$, there exists $\varepsilon > 0$ such that

$$\inf \mathbb{P}\left\{\inf_{d_g(\theta^*, \theta)\ge\delta}\left|\mathcal{L}(\theta) - \mathcal{L}(\theta^*)\right| \ge \varepsilon\right\} = 1,$$

4. There exist constants $(G, L)$ and a function $\mathcal{K}(x)$ such that for all $\theta, \theta' \in U$ and $t \in [0, 1]$

$$\mathbb{E}\|\nabla\mathcal{L}(\theta, \mathcal{D})\|^{16} \le G^{16}, \qquad \mathbb{E}\|\|\nabla^2\mathcal{L}(\theta, \mathcal{D}) - I(\theta)\|\|^{16} \le L^{16},$$

$$\|\nabla D(t, \theta, \theta')^*\nabla\mathcal{L}(\mathcal{R}_{\theta'}t\mathcal{R}_{\theta'}\theta, x)\| \le \mathcal{K}(x)d_{\mathcal{R}}(\theta, \theta'),$$

$$\left\|\left|\left(D\mathcal{R}_\theta^{-1}\hat{\theta}\right)^*\nabla^2\mathcal{L}(\theta, x)\left(D\mathcal{R}_{\hat{\theta}}^{-1}\theta\right)^{-1} - \left(D\mathcal{R}_{\theta'}^{-1}\hat{\theta}\right)^*\nabla^2\mathcal{L}(\theta', x)\left(D\mathcal{R}_{\hat{\theta}}^{-1}\theta'\right)^{-1}\right|\right\| \le \mathcal{K}(x)d_{\mathcal{R}_{\hat{\theta}}}(\theta, \theta'),$$

$$\left\|\left|\left(D\mathcal{R}_\theta^{-1}\hat{\theta}\right)^*\nabla^2\mathcal{L}(\theta, x)(D\mathcal{R}_\theta^{-1}\hat{\theta}) - \left(D\mathcal{R}_{\theta'}^{-1}\hat{\theta}\right)^*\nabla^2\mathcal{L}(\theta', x)\left(D\mathcal{R}_{\theta'}^{-1}\hat{\theta}\right)\right|\right\| \le \mathcal{K}(x)d_{\mathcal{R}_{\hat{\theta}}}(\theta, \theta'),$$

    where $\|\|\|$ is a spectral norm of matrices, $\|\|A\|\| = \sup\{\|Ax\| : x \in \mathbb{R}^n, \quad \|x\| = 1\}$. Moreover, $\mathcal{K}$ satisfies $\mathbb{E}\mathcal{K} \le K^{16}$ for some constant $K > 0$.

Given these conditions, we have the following theorem:

**Theorem 2.** If the standard regularity conditions holds, the initial estimator $\bar{\theta}$ lies in the neighborhood $U$ of $\theta^*$ and

$$\left\|\left(D\mathcal{R}_{\theta^*}^{-1}\hat{\theta}\right)^*\left(\nabla^2\tilde{\mathcal{L}}_s(\theta^*)-I(\theta^*)\right)\left(D\mathcal{R}_{\theta^*}^{-1}\hat{\theta}\right)\right\|\leq\frac{\rho\mu_-R_-}{4},$$

$$\left\|\left(D\mathcal{R}_\theta^{-1}\hat{\theta}\right)^*\nabla^2\tilde{\mathcal{L}}_s(\theta,x)\left(D\mathcal{R}_\theta^{-1}\hat{\theta}\right)-\left(D\mathcal{R}_{\theta'}^{-1}\hat{\theta}\right)^*\nabla^2\tilde{\mathcal{L}}_s(\theta',x)\left(D\mathcal{R}_{\theta'}^{-1}\hat{\theta}\right)\right\|\leq\mathcal{K}(x)d_{\mathcal{R}_{\hat{\theta}}}(\theta,\theta'),$$

where $R_-=\dfrac{1}{\left\|\left(\left(D\mathcal{R}_{\theta^*}^{-1}\hat{\theta}\right)^*\left(D\mathcal{R}_{\theta^*}^{-1}\hat{\theta}\right)\right)^{-1}\right\|}$, then any minimizer $\tilde{\theta}$ of the surrogate loss function $\tilde{\mathcal{L}}_s(\theta)$ satisfies

$$d_{\mathcal{R}}(\tilde{\theta},\hat{\theta})\leq C_2\Big(1+d_{\mathcal{R}}(\bar{\theta},\hat{\theta})+d_{\mathcal{R}}(\theta^*,\hat{\theta})+$$

$$C_3\left\|\left(D\mathcal{R}_{\theta^*}^{-1}\hat{\theta}\right)^*\left(\nabla^2\mathcal{L}_s(\theta^*)-\nabla^2\mathcal{L}_N(\theta^*)\right)\left(D\mathcal{R}_{\hat{\theta}}^{-1}\theta^*\right)^{-1}\right\|\Big)d_{\mathcal{R}}(\bar{\theta},\hat{\theta}),$$

with probability at least $1-C_1mn^{-8}$, where constants $C_1,C_2$ and $C_3$ are independent of $(m,n,N)$.

## 5  Simulation study and data analysis

To examine the quality of our parallel algorithm we first apply it to the estimation of Fréchet means on spheres, which has closed form expressions for the estimation of the extrinsic mean (true empirical minimizer). In addition, we apply our algorithm to Netflix movie-ranking data set as an example of optimization over Grassmannian manifolds in the low-rank matrix completion problem. In the following results, we demonstrate the utility of our algorithm both for high dimensional manifold-valued data (Section 5.1) and Euclidean space data with non-Euclidean parameters (Section 5.2). We wrote the code for our implementations in Python and carried out the parallelization of the code through MPI[1][7].

### 5.1  Estimation of Fréchet means on manifolds

We first consider the estimation problem of Fréchet means [10] on manifolds. In particular, the manifold under consideration is the sphere in which we wish to estimate both the extrinsic and intrinsic mean [3]. Let $M$ be a general manifold and $\rho$ be a distance on $M$ which can be an intrinsic distance, by employing a Riemannian structure of $M$, or an extrinsic distance, via some embedding $J$ onto some Euclidean space. Also, let $x_1,\ldots,x_N$ be sample of point on the hypersphere $S^d$, the sample Fréchet mean of $x_1,\ldots,x_n$ is defined as

$$\hat{\theta}=\arg\min_{\theta\in M=S^d}\sum_{i=1}^N\rho^2(\theta,x_i),\tag{2}$$

where $\rho$ is some distance on the sphere.

The extrinsic distance, for our spherical example, is defined to be $\rho(x,y)=\|J(x)-J(y)\|=\|x-y\|$ with $\|\cdot\|$ as the Euclidean distance and the embedding map $J(x)=x\in\mathbb{R}^{d+1}$ as the identity map. We call $\hat{\theta}$ the extrinsic Fréchet mean on the sphere. We choose this example in our simulation, as we know the true global optimizer which is given by $\bar{x}/\|\bar{x}\|$ where $\bar{x}$ is the standard sample mean of $x_1,\ldots,x_N$ in Euclidean distance. The intrinsic Fréchet mean, on the other hand, is defined to be where the distance $\rho$ is the geodesic distance (or the arc length). In this case we compare the estimator obtained from the parallel algorithm with the optimizer obtained from a gradient descent algorithm along the sphere applied to the entire data set. Despite that the spherical case may be an "easy" setting as it has a Betti number of zero, we chose this example so that we have ground truth to compare our results with and we, in fact, perform favorably even when the dimensionality of the data is high even as we increase the number of processors.

For this example, we simulate one million observations from a 100-dimensional von Mises distribution projected onto the unit sphere with mean sampled randomly from $N(0,I)$ and a precision of 2. For

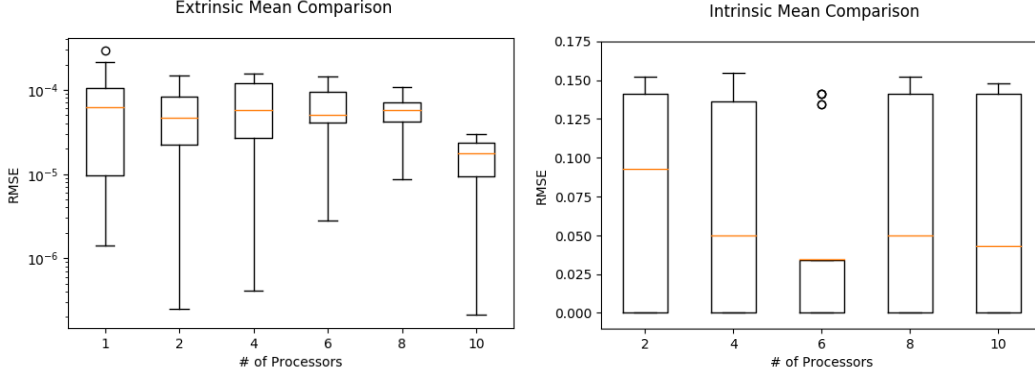

Figure 1: Extrinsic mean comparison (left) and intrinsic mean comparison (right) on spheres in $S^{99}$

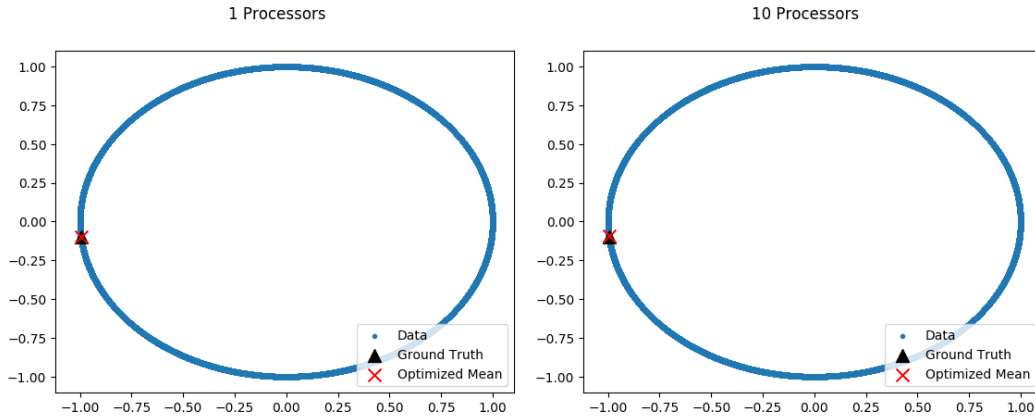

Figure 2: Extrinsic mean results on $S^1$, for one (left) and ten (right) processors

the extrinsic mean example, the closed form expression of the sample mean acts as a "ground truth" to which we can compare our results. In both the extrinsic and intrinsic mean examples, we run 20 trials of our algorithm over 1, 2, 4, 6, 8 and 10 processors. For the extrinsic mean simulations we compare our results to the true global optimizer in terms of root mean squared error (RMSE) and for the intrinsic mean simulations we compare our distributed results to the single processor results, also in terms of RMSE.

As we can see in Figure 1, even if we divide our observations to as many as 10 processors we still obtain favorable results for the estimation of the Fréchet mean in terms of RMSE to the ground truth for the extrinsic mean case and the single processor results for the intrinsic mean case. To visualize this comparison, we show in Figure 2 an example of our method's performance on two dimensional data so that we may see that our optimization results yield a very close estimate to the true global optimizer.

## 5.2   Real data analysis: the Netflix example

Next, we consider an application of our algorithm to the Netflix movie rating dataset. This dataset of over a million entries, $X \in \mathbb{R}^{M \times N}$, consists of $M = 17770$ movies and $N = 480189$ users, in which only a sparse subset of the users and movies have ratings. In order to build a better recommendation systems to users, we can frame the problem of predicting users' ratings for movies as a low-rank matrix completion problem by learning the rank-$r$ Grassmannian manifold $U \in \text{Gr}(M, r)$ which optimizes for the set of observed entries $(i, j) \in \Omega$ the loss function

$$L(U) = \frac{1}{2} \sum_{(i,j) \in \Omega} \left( (UW)_{ij} - X_{ij} \right)^2 + \frac{\lambda^2}{2} \sum_{(i,j) \notin \Omega} (UW)_{ij}, \tag{3}$$

where $W$ is $r$-by-$N$ matrix. Each user $k$ has the loss function $\mathscr{L}(U,k) = \frac{1}{2}|c_k \circ (U w_k(U) - X_k)|^2$, where $\circ$ is the Hadamard product, $(w_k)^i = W_{ik}$, and

$$(c_k)^i = \begin{cases} 1, & \text{if } (i,k) \in \Omega \\ \lambda, & \text{if } (i,k) \notin \Omega \end{cases}, \qquad (X_k)^i = \begin{cases} X_{ik}, & \text{if } (i,k) \in \Omega \\ 0, & \text{if } (i,k) \notin \Omega, \end{cases}$$

$$w_k(U) = \left(U^T \text{diag}(c_k \circ c_k) U\right)^{-1} U^T \left(c_k \circ c_k \circ X_k\right).$$

Which results in the following gradient

$$\nabla \mathscr{L}(U,k) = \left(c_k \circ c_k \circ (U w_k(U) - X_k)\right) w_k(U)^T = \text{diag}(c_k \circ c_k)(U w_k(U) - X_k) w_k(U)^T.$$

We can assume that $N = pq$, then for each local machine $\mathscr{M}_j$, $j = 1, ..., p$, we have the local function $\mathscr{L}_j(U) = \frac{1}{q} \sum_{k=(j-1)q+1}^{jq} \mathscr{L}(U,k)$. So the global function is

$$\mathscr{L}_N(U) = \frac{1}{p} \sum_{j=1}^{p} \mathscr{L}_j(U) = \frac{1}{pq} \sum_{k=1}^{pq} \mathscr{L}(U,k) = \frac{1}{N} L(U).$$

For iterations $s = 0, 1, ..., P-1$ we have $\nabla \mathscr{L}_j(U_s) = \sum_{k=(j-1)q+1}^{jq} \nabla \mathscr{L}(U_s, k)$. Therefore the global gradient is $\nabla \mathscr{L}_N(U_s) = \frac{1}{p} \sum_{j=1}^{p} \nabla \mathscr{L}_j(U_s)$. Instead of the logarithm map we will use the inverse retraction map

$$\begin{aligned} \mathrm{R}_{[U]}^{-1}: \quad \mathrm{Gr}(m,r) \quad &\to \quad T_{[U]}\mathrm{Gr}(m,r) \\ [V] \quad &\mapsto \quad V - U(U^T U)^{-1} U^T V. \end{aligned}$$

Which gives us the following surrogate function

$$\begin{aligned} \tilde{\mathscr{L}}_s(V) &= \mathscr{L}_s(V) - \langle V - U_s(U_s^T U_s)^{-1} U_s^T V, \nabla \mathscr{L}_s(U_s) - \nabla \mathscr{L}_N(U_s) \rangle \\ &= \mathscr{L}_s(V) - \langle V, \nabla \mathscr{L}_s(U_s) - \nabla \mathscr{L}_N(U_s) \rangle. \end{aligned}$$

and its gradient

$$\nabla \tilde{\mathscr{L}}_s(V) = \nabla \mathscr{L}_s(V) - (I_m - V(V^T V)^{-1} V^T)(\nabla \mathscr{L}_s(U_s) - \nabla \mathscr{L}_N(U_s)).$$

To optimize with respect to our loss function, we have to find $U_{s+1} = \text{argmin}\, \tilde{\mathscr{L}}_s$. To do this, we move according to the steepest descent by taking step size $\lambda_0$ in the direction $\nabla \tilde{\mathscr{L}}_s(U_s)$ by taking the retraction, $U_{s+1} = R_{[U_s]}\left(\lambda_0 \nabla \tilde{\mathscr{L}}_s(U_s)\right)$.[2]

For our example we set the matrix rank to $r = 10$ and the regularization parameter to $\lambda = 0.1$ and divided the data randomly across 4 processors. Figure 3 shows that we can perform distributed manifold gradient descent in this complicated problem and we can reach convergence fairly quickly (after about 1000 seconds).

## 6  Conclusion

We propose in this paper a communication efficient parallel algorithm for general optimization problems on manifolds which is applicable to many different manifold spaces and loss functions. Moreover, our proposed algorithm can explore the geometry of the underlying space efficiently and perform well in simulation studies and practical examples all while having theoretical convergence guarantees.

In the age of "big data", the need for distributable inference algorithms is crucial as we cannot reliably expect entire datasets to sit on a single processor anymore. Despite this, much of the previous work in parallel inference has only focused on data and parameters in Euclidean space. Realistically, much of the data that we are interested in is better modeled by manifolds and thus we need fast inference algorithms that are provably suitable for situations beyond the Euclidean setting. In future work, we aim to extend the situations under which parallel inference algorithms are generalizable to manifolds and demonstrate more critical problems (in neuroscience or computer vision, for example) in which parallel inference is a crucial solution.

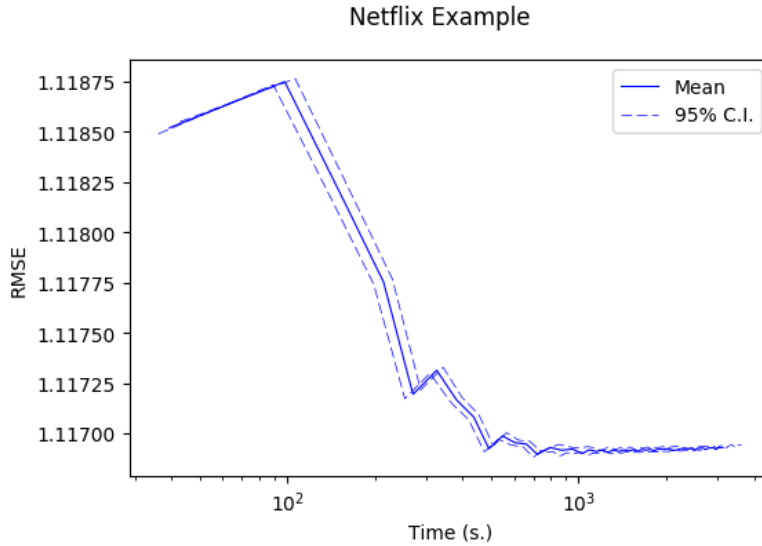

Figure 3: Test set RMSE of the Netflix example over time, evaluated on 10 trials.
.

**Acknowledgments**

Bayan Saparbayeva was partially supported by DARPA N66001-17-1-4041. Michael Zhang was supported by NSF grant 1447721. Lizhen Lin acknowledges the support from NSF grants IIS 1663870, DMS Career 1654579 and a DARPA grant N66001-17-1-4041.

## Footnotes

[1]Our code is available at `https://github.com/michaelzhang01/parallel_manifold_opt`

[2]We select the step size parameter according to the modified Armijo algorithm seen in [6].

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
