[Reviews · NeurIPS 2018]

Reviewer 1



This paper develops parallel algorithms in a manifold setting. For doing this, this paper uses Iterative Local Estimation Algorithm with exponential map and logarithm map on a manifold. They show the upper bound for the loss under some regularity conditions. They also verify their algorithm on simulated data and real data. In terms of quality, their method is both theoretically and experimentally verified. Their theoretical analysis is sound with the concrete assumptions and bound for the loss. In terms of clarity, this paper is clearly written and well organized. Introduction gives good motivation for why parallelization on optimization algorithm on a manifold is important. In terms of originality, adapting parallel inference framework in a manifold setting is original and new as I know. References for parallel inference framework are adequately referenced. In terms of significance, their method enables to parallelize inference on manifolds, which is in general difficult. Their theoretical analysis gives an upper bound for the loss with high probability. And their experimental results also validate the correctness of their algorithms. I, as the reviewer, am familiar to differential geometry on manifolds but not familiar to parallel optimizations. And I also have some minor suggestions in submission: Submission: 66th line: D is the parameter dimensionality: shouldn't D be the data space? 81th line: 2nd line of the equation is not correct. It should be 1/2 < theta-bar{theta}, (Hessian L_1(bar{theta}) - Hessian L_N(bar{theta})) (theta-bar{theta}) > or something similar to this. 118th line: In R_{bar{theta}_1} t R_{bar{theta}_1} theta_2, bar should be removed. 123th line: log_{bar{theta}}^{-1} theta -> log_{bar{theta}} theta 146th line: L' should be appeared before the equation, something like "we also demand that there exists L' in R with (math equation)" 229th line: aim extend -> aim to extend ------------------------------------------------------------------------------- Comments after Author's Feedback I agree to Reviewer 4 to the point that the authors need to provide better motivations for how the process communication becomes expensive in for the optimization on manifolds. But I am convinced with the author's feedback about the motivation and still appreciates their theoretical analysis of convergence rates, so I would maintain my score.

Reviewer 2



1. In line 113, what is R_{\theta}? 2. In line 117, it should be triangle inequality not triangular. 3. In line 126, addictive--> additive. 4. Inverse retraction map is called lifting. Also, please mention the radius of the regular geodesic ball inside which retraction map is a bijection. 5. In line 169, please bold x_is and bold S in hypersphere. 6. Eq. (2) is FM iff \rho is geodesic distance. 7. In line 177, what is 'big circle distance', should be arc length. 8. The fastest state-of-the-art FM estimator algorithm on hypersphere is by Chakraborty et al. in MFCA 2015. Please comment on performance improvement compare to this online algorithm. Overall, though the idea is nice, I believe it mostly depends on the previous theorem (Thm. 1). Experimental section lacks comparison with the state-of-the-art. Needs proof reading and significant corrections with notations, technical content. ---------------------------------------------------------------------------------------------- POST REBUTTAL COMMENTS: The authors did a decent job with the rebuttal. My comments are successfully addresses (more-or-less). I was though concern about the motivation after reading R4's reviews, my opinion is authors did a good job there as well. I agree that minimizing the exact loss function is not feasible, hence the solution proposed is well motivated after reading the rebuttal. So, in summary, I vote for acceptance and am going to increase my vote to a 7 assuming that authors will do a check for correctness of the technical content, notations etc..

Reviewer 3



The rebuttal letter of the authors carefully considered my remarks, committed to fix some of the issues, while deferred some more substantial work to the future. I am pleased to accept the manuscript after having read the perceptive rebuttal letter, which addressed most of my considerations. To me, the most fundamental issue is that some of the assumptions made in the paper don't hold in contemporary distributed computing systems, but I am open to accept the paper in terms of its theoretical contribution. This paper builds upon and extends the iterative local estimation algorithm (ILEA) in two ways. Firstly, this paper allows optimization to take place on a different machine between iterations. Secondly, the more general so-called retraction map is used in place of the exponential map. Ultimately, the goal of the paper is to scale global optimization in a parallel fashion while exploiting the geometry of the parameter space. The paper aims at addressing the important problem of scalable parallel optimization and takes reasonable steps towards this aim. At the same time, there are several points to consider attentively towards the proclaimed aim: 1.1) It is not entirely clear to me for which parallel architecture the algorithm is designed. Is the goal to use the algorithm for high performance computing or for distributed or cloud computing? On the basis of lines 31-33, it seems to me that the authors have designed the algorithm for low latency distributed frameworks, such as Hadoop or Spark. If this is the case, low latency is not the only design principle for a scalable distributed algorithm. It is also important that the algorithm is designed keeping in mind that it is meant to work on data management layers, such as HDFS and YARN. Thus, the assumption made in lines 63-65 about IID observations across different machines is not aligned with the reality of batch and real-time data workloads provided by YARN. In more statistical terms, online learning problems can't be based on IID distributed data across nodes, and in general the users, developers or engineers don't manage cloud computing at so low-level so as to align data allocation according to such an IID assumption through time. 1.2) Along the lines of the above remark, it is not mentioned in the examples which parallel framework was used for running the proposed algorithm. Did you use an implementation of the message passing interface (MPI) or the actor model in Akka for instance? You may provide more information about these matters in section 5 (simulation study). 1.3) The first example is based on a sphere. Given that the Betti number of a sphere is zero, this example does not provide a geometrically challenging parameter space. It would be useful to use a more challenging manifold. 1.4) The algorithm is also ran on Netflix data. The issue here is that there is no absolute RMSE to compare the proposed algorithm to other established optimization methods. The landmark paper on optimization 'Lipschitzian optimization without the Lipschitz constant' by Jones, Perttunen and Stuckman, journal of optimization theory and application, 1993, provides nine test functions for benchmarking optimization algorithms, see Table 1 and equation (12) in Jones' paper. To get an idea of how well your proposed algorithm works, I suggest you run it on these benchmark functions and you tabulate the results along with the analogous results for other standard optimization methods. 1.5) You mention in lines 35-36 that your goal is to address challenges arising from inference on big non-Euclidean data or data with non-Euclidean parameters. Since you define the manifold M in the parameter space in lines 104-105, my understanding is that the paper addresses only the latter case (data with non-Euclidean parameters), so I would suggest you drop the reference to the former case (big non-Euclidean data) from lines 35-36. Some minor fixes are proposed below: 2.1) Line 18: Change 'include diffusions matrices' to 'include diffusion matrices'. 2.2) Line 30: A useful paper missing from the list of relevant references you provide in line 30 is the following, 'Merging MCMC subposteriors through Gaussian-process approximations' by Nemeth and Sherlock. 2.3) Line 66: It is mentioned that calligraphic D refers to the parameter dimensionality. If not mistaken, calligraphic D refers to data rather than parameters in your paper? 2.4) Lines 71-72: You may want to rephrase this sentence, there seems to be unintended repetition of the point about communication. 2.5) Lines 104-105: The symbols d and M appear in lines 104-105 for the first time if not mistaken. State more explicitly here that d is the dimension of the parameter space and that the manifold M lives in the parameter space. It can be deduced by reading carefully, yet it would help the reader to state this more directly. 2.6) Line 117: The notation d_{R} has not been introduced. It can be guessed that d_{R} is a metric given that you refer to the triangular inequality, yet it would be helpful to introduce notation more explicitly. 2.7) Line 126: Change 'the addictive constant' to 'the additive constant'. 2.8) Lines 134-135: It is not clear to me what 'population risk' means. Do you mean 'global risk'? 2.9) From line 150 onwards, you use three vertical lines instead of two; do the three vertical lines refer to a norm different than the norm represented by two vertical lines? If this is not a typo, introduce the relevant notation (unless of course you have already explain this in the paper and it eluded me). 2.10) In equation (3), you introduce the symbol 'W'. Does W_{ij} denote a binary variable indicating the presence or absence of an edge between the i-th movie and j-th user? Introduce the notation W_{ij} and its possible relation with the symbol w_k appearing right before line 203.

Reviewer 4



The paper describes a technique for distributed computations in settings where the parameters are modeled to lie on a manifold. They derive convergence results for their proposed algorithm and evaluate it on a synthetic problem and the Netflix data. In my opinion, the big weakness of the paper is its insufficient motivation of the problem, and thus, I fail to see the relevance. Specifically, the authors do not show that the process communication is indeed a problem in practice. This is particularly surprising as typical empirical risk minimization (ERM) problems naturally decompose as sum of losses and the gradient is the sum of the local loss gradients. Thus, the required inter-process communication limits to one map-reduce and one broadcast per iteration. The paper lacks experiments reporting runtime vs. number of processors, in particular with respect to different communication strategies. In effect, I cannot follow _why_ the authors would like to proof convergence results for this setting, as I am not convinced _why_ this setting is preferable to other communication and optimization strategies. I am uncertain whether this is only a problem of presentation and not of significance, but it is unfortunately insufficient in either case. Here are some suggetions which I hope the authors might find useful for future presentations of the subject: - I did not get a clear picture from the goal of the paper in the introduction. My guess is that the examples chosen did not convince me that there are problems which require a lot of inter-process communication. This holds particularly for the second paragraph where sampling-based Bayesian methods are particularly mentioned as an example where the paper's results are irrelevant as they are already embarrassingly parallel. Instead, I suggest the authors try to focus on problems where the loss function does not decompose as the sum of sample losses and other ERM-based distributed algorithms such as Hogwild. - From the discussion in lines 60-74 (beginning of Sect. 2), I had the impression that the authors want to focus on a situation where the gradient of the sum is not the sum of the individual gradients, but this is not communicated in the text. In particular, the paragraph lines 70-74 is a setting that is shared in all ERM approaches and could be discussed in less space. A situation where the the gradient of the sum of the losses is not the sum of the individual loss gradients is rare and could require some space. - Line 114, the authors should introduce the notation for audiences not familiar with manifolds ('D' in "D R_theta ...") - From the presentations in the supplement, I cannot see how the consideration of the events immediately imply the Theorem. I assume this presentation is incomplete? Generally, the authors could point which parts explicitely of the proof need to be adapted, why, and how it's done. The authors also seem to refer to statements in Lemmata defined in other texts (Lemma 3, Lemma 4). These should be restated or more clearly referenced. - In algorithm 1, the subscript 's' denotes the master machine, but also iteration times. I propose to use subscript '1' to denote to the master machine (problematic lines are line 6 "Transmit the local ..." and lines 8-9 "Form the surrogate function ...") - the text should be checked for typos ("addictive constant" and others) --- Post-rebuttal update: I appreciate the author's feedback. I am afraid my missing point was on a deeper level than the authors anticipated: I fail to see in what situations the access to the global higher-derivatives is required which seems to be the crux and the motivation of the analysis. In particular, to me this is still in stark contrast with the rebuttal to comment 2. If I'm using gradient descent and if the gradient of the distributed sum is the sum of the distributed gradients, how is the implementation more costly than a map-reduce followed-up by a broadcast? For gradient descent to converge, no global higher-order derivative information is required, no? So this could be me not having a good overview of more elaborate optimization algorithms or some other misunderstanding of the paper. But I'm seeing that my colleagues seem to grasp the significance so I'm looking forward to be convinced in the future.